# Structural and Magnetic Specificities of Fe-B Thin Films Obtained by Thermionic Vacuum Arc and Magnetron Sputtering

**Cornel Staicu** [1,2,*], **Claudiu Locovei** [2,3], **Andrei Alexandru Dinu** [2,3], **Ion Burducea** [4], **Paul Dincă** [1], **Bogdan Butoi** [1], **Oana Gloria Pompilian** [1], **Corneliu Porosnicu** [1], **Cristian Petrica Lungu** [1] and **Victor Kuncser** [3,*]

[1] Low Temperature Plasma Physics Department, National Institute for Laser, Plasma and Radiation Physics, 077125 Magurele, Romania
[2] Faculty of Physics, University of Bucharest, 077125 Magurele, Romania
[3] Magnetism and Superconductivity Department, Institute of Materials Physics, 077125 Magurele, Romania
[4] Horia Hulubei National Institute of Physics and Nuclear Engineering, 077125 Magurele, Romania
[*] Correspondence: cornel.staicu@inflpr.ro (C.S.); kuncser@infim.ro (V.K.)

**Abstract:** Fe-B based compounds are of high interest due to their special properties and the wide range of involved applications. While B is the element that facilitates the increase in the hardness and the degree of wear resistance, it is also an effective glass former, controlling the formation of a much-desired amorphous structure with specific magnetic properties. Major difficulties related to the proper engineering of Fe-B thin films lay especially in their preparation under well-defined compositions, which in turn, should be accurately determined. The present study closely analyzes the morpho-structural and magnetic properties of thin coatings of Fe-B of approximately 100 nm thickness and with the nominal B content ranging from 5 at. % to 50 at. %. The comparison between films obtained by two preparation methods, namely, the thermionic vacuum arc and the magnetron sputtering is envisaged. Morpho-structural properties were highlighted using X-ray diffraction supplemented with X-ray reflectometry and scanning electron microscopy, whereas the elemental investigations were performed by X-ray dispersive spectroscopy and Rutherford back-scattering spectroscopy. The magnetic properties of the Fe-B layers were carefully investigated by the vectorial magneto-optic Kerr effect and conversion electron Mössbauer spectroscopy. The high capability of Mössbauer Spectroscopy to provide the phase composition and the B content in the formed Fe-B intermetallic films was proven, in correlation to Rutherford back-scattering techniques, and to explain their magnetic properties, including the magnetic texture of interest in many applications, in correlation with longitudinal magneto-optic-Kerr-effect-based techniques.

**Keywords:** TVA; magnetron Sputtering; Fe-B films; magnetic texture; Mössbauer spectroscopy

## 1. Introduction

Since the 1970s, when the preparation of amorphous Fe-P-C compounds used an ultrafast solidification technique [1], many other amorphous metals containing iron and appropriate glass formers have been developed. Such alloys present a particular interest, especially in the field of electrical engineering (transformers and sensors based on magneto-impedance effects), due to their special soft magnetic characteristics. In addition to P and C, B is also a good glass forming agent helping in the formation of Fe-B amorphous compounds with relatively low concentrations of B. Except in its combination with Fe, the unique properties of boron and its behavior in relation to other metals have led to the production of a large class of borons, from $YB_{66}$ (monochromator for synchrotron radiation), to $(MgB_2)$ [2], known for its superconducting properties, and $(AlFe_2B_2)$ [3], known for its magneto-caloric properties. Last but not least, boron is essential in the class of $Nd_2Fe_{14}B$

hard magnetic materials [4–8]. In literature, the interest in iron boride is already well known. Iron–boron compounds are of considerable interest because boride layers on iron and steel have a high wear resistance [9–13]. Iron borons are also used as deoxidizing agents, designed to increase the hardness of steel and coating alloy components applied to welded electrodes. However, despite the widespread application of these borons in industry, their physical properties have not been sufficiently investigated. Although Fe-B based metastable and amorphous systems can be easily obtained by melt spinning methods as ribbons for a narrow range of B content [14,15], more challenging preparation and investigation issues appear in the case of the counterpart thin films. Only a few studies are reported on the magnetic properties of Fe-B based amorphous intermetallic thin films, e.g., containing also rare earth (RE) elements [16]. In the case of thin films, strong variations of both mechanical and magnetic properties are expected versus the preparation conditions, the involved techniques, as well as the B content, all these issues defining finally whether the films are amorphous or not. The accurate determination of not only the average B concentration in the film but also of its distribution throughout the film are other major challenges, as many of the basic techniques are not able to provide reasonable elemental analysis for the light elements (including B).

In light of the above, the present paper aims to make a comparison between the structural and magnetic properties of thin coatings of Fe-B with various concentrations, as obtained by two deposition methods: (i) thermionic vacuum arc (TVA) and (ii) magnetron sputtering (MS). To determine the structural and compositional properties, the obtained thin films were investigated using X-ray diffraction (XRD), X-ray reflectometry (XRR), scanning electron microscopy (SEM), energy dispersion X-ray spectroscopy (EDX), and Rutherford backscattering (RBS) spectroscopy. At the same time, the magnetic properties of the Fe-B layers were carefully investigated by vectorial magneto-optic Kerr effect (MOKE) and conversion electron Mössbauer (CEM) spectroscopy.

## 2. Experimental Details

Fe-B thin films of different concentrations were prepared by the two methods mentioned above. The nominal concentrations, presented in Table 1, second column, correspond to the imposed effective thicknesses of Fe and B presented in columns 3 and 4. Accordingly, the specific deposition rates of the two elements were calculated and controlled via quartz microbalance measurements used in the cases of each deposition method (TVA and MS, respectively).

**Table 1.** Prepared samples, their nominal composition and corresponding effective thickness for each element.

| No. | Concentration (at. %) | Fe (nm) | B (nm) |
|-----|----------------------|---------|--------|
| 1 | Fe + B (5%) | 96.9 | 3.3 |
| 2 | Fe + B (10%) | 97.3 | 7.0 |
| 3 | Fe + B (20%) | 86.5 | 14.0 |
| 4 | Fe + B (30%) | 79.3 | 22.0 |
| 5 | Fe + B (40%) | 70.0 | 30.0 |
| 6 | Fe + B (50%) | 60.0 | 40.0 |

### 2.1. Preparation by Thermionic Vacuum Arc versus Magnetron Sputtering

To obtain thin films with well-defined atomic compositions (from 5 at. % to 50 at.% of B) by simultaneous deposition of Fe and B, two specific anode–cathode systems were used in case of TVA. They consist of a heated tungsten filament and a heat-resistant anode made of either titanium diboride ($TiB_2$) for boron or of carbon for iron (see Figure 1) [17]. The electrons emitted by the incandescent filament of W are subsequently accelerated to the anode by a Wehnelt cylinder acting as an electrostatic lens by applying a positive voltage [18–20]. In the first step, the material of interest is heated by electron bombardment. The second part consists of melting the material, giving rise to a pressure of metal vapors. Subsequently, due

to the inelastic collisions between the thermally emitted filament electrons and the Fe and B atoms, a plasma ignition results in the pure metallic vapors of these materials. By carefully controlling the plasma parameters (filament current, discharge current, respectively, the voltage applied on the anode) a constant evaporation rate is obtained for each material. The main advantage of the TVA method with respect to MS (draws of the two discharge configurations are presented in Figure 1) is the high quality of the deposits due to the high vacuum conditions (lack of an additional discharge gas, e.g., Ar). Films characterized by a high degree of purity are formed due to the absence of gaseous inclusions [21]. Another advantage is the compactness of the coating with higher adhesion to the substrate.

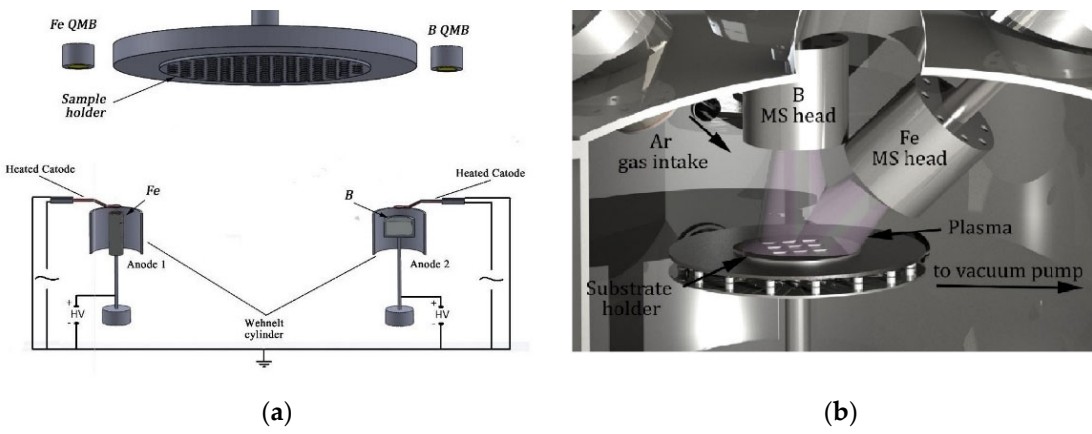

(**a**)  (**b**)

**Figure 1.** Experimental set-up for Fe-B deposits by (**a**) TVA method (left hand) and by (**b**) MS (right hand), respectively.

The second method used to obtain Fe-B thin films is MS. For this experiment, Fe-B deposits were made using r.f. magnetron sputtering for B and the dc sputtering mode for Fe. The substrates were mounted in the vacuum chamber on a fixed circular holder placed 10 cm from the sputtering source. The used magnetron system was composed of a single water-cooled cathode, provided with two circular targets of B and Fe (2 mm thick and 50 mm in diameter each). To clean the targets of oxides and other impurities, an argon discharge plasma was initially ignited for approximately 30 min. Throughout the cleaning process, the sample holder was shielded from the plasma with a shutter. The working pressure of Ar was maintained at $1.5 \times 10^{-2}$ under a continuous Ar flow of 20 sccm. An in situ acceleration voltage of $-60$ V was also used on the substrates to improve the adhesion and compactness of the deposited layers.

The substrates for the coatings in this study, independent of the deposition method, consisted of silicon wafers with a 12 mm × 15 mm size, with an optically polished surface and an average roughness of 6 nm. The substrates mounted in the vacuum chamber were ultrasonically cleaned using a solution of acetone as the first cleaning agent and isopropyl alcohol in the second stage and finally dried in atmospheric conditions. Fe-B thin films with a thickness of 100 nm were obtained under real time monitoring conditions via quartz microbalance systems. The base pressure was better than $10^{-5}$ mbar in each case.

## 2.2. Characterization

The structural and morphological properties of the obtained samples were analyzed using XRD, XRR and SEM, whereas the compositional measurements were performed by EDX and RBS.

XRD measurements were performed in Bragg–Brentano θ-2θ geometry at room temperature using the Bruker D8 ADVANCE diffractometer (CuKα = 1.54 Å) (Bruker Scientific Instruments, Madison, IA, USA). The data acquired for the determination of the crystalline phases were recorded in the range 2θ = 35–60° at a scanning rate equal to 0.02°/s. Film uniformity and thickness along with Fe-B thin film density were investigated by the

same device working under the XRR mode with a scan rate of 0.008°/s. The Z-alignment procedure was initially performed, and the flatness of the sample was calibrated by the maximum of rocking curve.

For the SEM measurements, a scanning electron microscope, FEI Co., Inspect S. model (LabX Media Group, Midland, ON, Canada) was used, with the electron acceleration voltage varying between 0–30 kV to a working distance in the range of 0–30 mm under high vacuum conditions. The parameters used for the measurements were: acceleration voltage set to 25 kV, magnification 10,000X, working distance 13.5 mm, pressure of $10^{-4}$ Pa. Energy-dispersive X-ray spectroscopy (EDX) (LabX Media Group, Midland, ON, Canada) was carried out for topographic inspection and local chemical analysis. Typically, it is impossible to detect B by EDX. However, O X-ray yields are easily identified. The INCA package software (version 7.3) was used for the EDX elemental analysis. The energy calibration of the EDX facility was carried out with the K$\alpha$ X-ray emissions of copper (8.05 keV) and of silicon (1.74 keV).

For RBS measurements, the samples were mounted on a goniometric support adjustable on 3 axes with an accuracy of 0.01°. The solid silicon detector with an energy resolution of 18 keV used for the detection of retro-scattered particles of He was positioned at an angle of 165° with respect to the ion beam. A 0.5 mm$^2$ wide beam of He 2.6 MeV monoenergetic ions was focused on the sample. In addition to RBS, the resonance method, non-Rutherford backscattering spectrometry (NRBS) (High Voltage Engineering Europe B.V., Amsterdamseweg, Netherlands), using 3.042, 3.884 and 4.282 MeV He beams, was also applied for crosschecking the stoichiometry and thickness of the samples.

The magnetic characterizations were performed at room temperature (RT) via a vectorial MOKE device (M&A) (Quantum Design, San Diego, CA, USA) which allows the acquisition of hysteresis loops for various orientations of the applied magnetic field with respect to a fixed direction in the film plane. The in-plane angular distribution of the magnetization easy axis becomes accessible with direct indications on the magnetic texture of the films. Finally, microscopic aspects related to magnetic behavior and local interactions in relation to atomic and electronic configurations around the Fe atoms were analyzed by RT CEM spectroscopy. A $^{57}$Co (Rh) radioactive source was installed on a constant acceleration spectrometer. The sample was introduced in a gas flow proportional counter working with a mixture of methane (5%) and He. The $\gamma$ radiation was incident to the sample plane, and the fitting of the Mössbauer spectra was achieved through the NORMOS software (version 1.0.11) [22]. Isomer shift values were reported relative to metallic $\alpha$-Fe at RT.

## 3. Results and Discussion

The results will be comparatively presented on samples obtained using the two deposition methods, in terms of structure and morphology, as well as of the magnetic properties.

### 3.1. Morphology and Structure of Fe-B Films

The comparative SEM measurements on similar films from the two batches of samples (obtained by TVA and MS, respectively) are presented in Figure 2. Films from the two batches, with 20 at. % and 50 at. % of B, were chosen to be presented due to the clear differences between each other.

Figure 2 shows the SEM images of Fe-B films at a magnification of 10 kX. The morphology is lacking significant roughness on the surface of the films in both deposition batches. However, a clear difference can be seen between the films obtained by TVA and MS. According to Figure 2, the level of mesoscopic uniformity and roughness of the films obtained by TVA is clearly superior to the ones obtained by MS. Lower differences (films of very good uniformity) were observed for other B contents.

XRD patterns on two samples from each batch are presented for comparison in Figure 3.

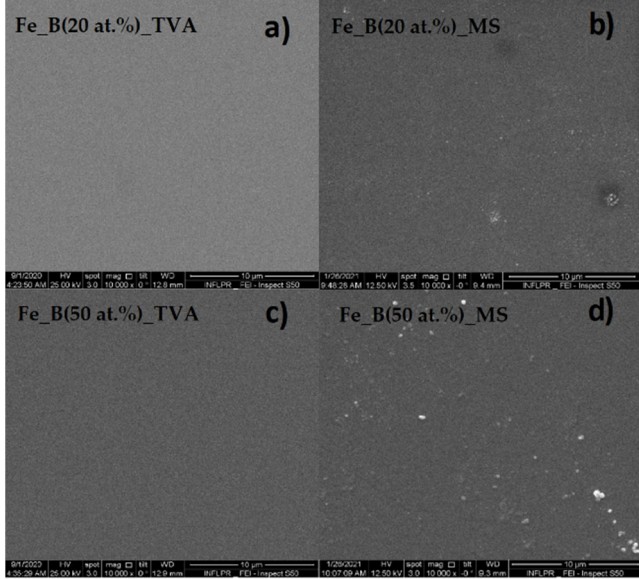

**Figure 2.** SEM images for samples: (**a**) Fe-B (20 at. % of B) obtained by TVA, (**b**) Fe-B (20 at. % of B) obtained by MS, (**c**) Fe-B (50 at. % of B) obtained by TVA, (**d**) Fe-B (50 at. % of B) obtained by MS.

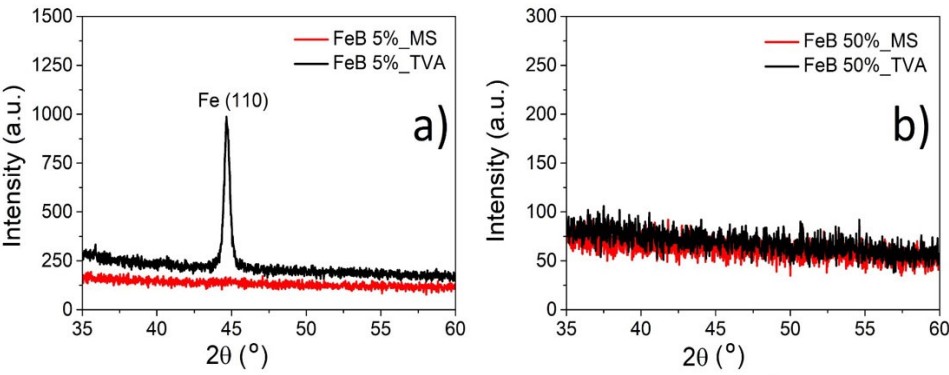

**Figure 3.** XRD patterns of Fe-B thin films with 5 at. % of B (**a**) and 50 at. % of B (**b**).

First, it is observed that for low concentrations of B (5%), a well crystallized film is obtained by TVA (black line), whereas an amorphous-like or nanocrystalline film (red line) is obtained by the more energetic MS method. For high concentrations of B, the films obtained by both methods are amorphous. However, XRD is not able to determine an accurate estimation of the relative content of crystalline and amorphous phase, as expected at low to intermediate content of B. In this respect, CEM spectroscopy results, as presented subsequently, are more concluding.

The XRR measurements for both films with 20 at. % boron nominal composition obtained by TVA and MS, respectively, are shown in Figure 4. Much better formed oscillations are evidenced in the XRR pattern of the film obtained by TVA, as direct proof of a much better uniformity of the film at nanoscale compared to the case of the similar film obtained by MS. However, from the involved beats, a different thickness is evidenced for the two films: only approximately 71(1) nm in case of the film obtained by TVA and 103(2) nm for the film obtained by MS.

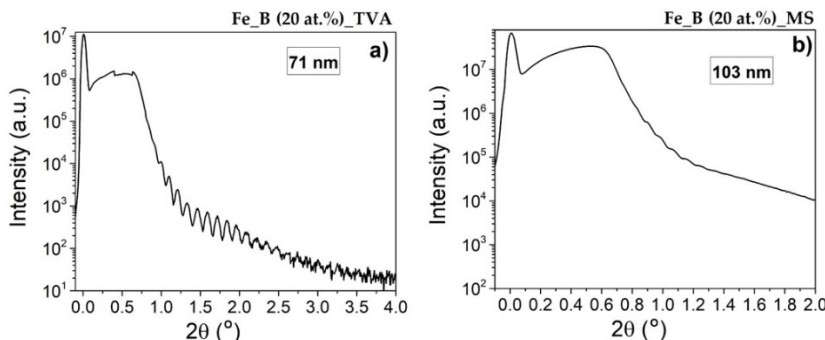

**Figure 4.** XRR patterns of Fe-B films (20 at. % B) obtained by TVA (**a**) and MS (**b**).

EDX measurements were made on the two films with 50 at. % of B obtained by TVA and MS, respectively. The difficulty of an accurate determination of light elements (e.g., Be, B) by this technique is well known. However, the EDX spectra of the two samples with the highest nominal concentrations of B (50 at. %) prepared by the two methods give evidence for a lower amount of residual oxygen and to a higher boron content in the sample prepared by TVA as compared to the one prepared by MS, and consequently, to a much more proper ratio between Fe and B in the first sample. Nevertheless, this is rather a qualitative result and not an accurate one, especially related to the B amount. A much more proper method for the elemental investigation of such films is RBS. The elemental content was also checked by NRMS, which is even more sensitive to the light elements when compared to RBS, especially when the substrate has a high Z number.

An exemplification of the RBS data for samples of different B content obtained by MS are presented in Figure 5a, whereas NRBS spectra collected on samples obtained by TVA are shown in Figure 5b. The RBS spectra were simulated using SIMNRA software package [23]. Areal density or thin film units ($10^{15}$ atoms/cm$^2$) are the natural units for ion beam analysis (IBA) since the energy loss is measured in eV/(atoms/cm$^2$), and one monolayer is of the order of $10^{15}$ atoms/cm$^2$ [24,25]. The following tables present the RBS results regarding the stoichiometry of the samples.

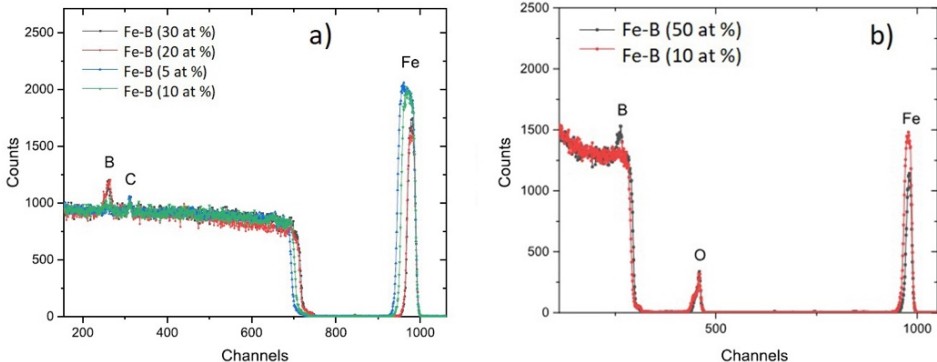

**Figure 5.** RBS data for Fe-B samples of different B concentrations obtained by MS (**a**) and NRBS spectra for samples obtained by TVA (**b**).

On all the samples, the presence of some impurities that could be due to some surface contamination of the samples is noticeable, e.g., it is well-known that also during ion beam analysis there is a carbon build-up on the sample surface. This is a mixture of C, H and O with thicknesses in the order of 50 TFU (1 TFU = $10^{15}$ atoms/cm$^2$). Other contaminants may be present on the sample surfaces (in a very low concentration) due to the handling or from some other reasons. In addition, a small and negligible concentration of W of 0.03% was observed in all TVA samples, probably resulting from the tungsten filament used as a cathode in TVA deposition. However, the most significant observation is related to an order of magnitude higher content of oxygen (tenths of at. %) in samples obtained

by MS compared to samples obtained by TVA (order of at. %) showing definitively the much better quality of samples obtained by the second method. Concerning the Fe-B films obtained by MS, the Mössbauer results show the presence of an Fe oxide phase superposed over Fe-B intermetallic phases, and therefore, the average composition of the Fe-B phase cannot be subtracted directly from Table 2 by a simple renormalization between Fe and B. We noted that the $^{57}$Fe Mössbauer results presented in the next section definitively prove the lack of Fe oxide phases and the formation of only Fe-B intermetallic phases in the Fe-B samples obtained by TVA. In this case, the few at. % of oxygen in the samples should be seen as inclusions and not making chemical bounds. As a consequence, the oxygen content will influence only the relative content of Fe and B in the Fe-B phases formed in the films, with direct influence on their magnetic properties. The real elemental contents in the Fe-B thin films obtained by TVA (in at. %), to be considered in the following for the interpretation of Mössbauer and magnetic results are presented in Table 3.

**Table 2.** Atomic composition of samples Fe-B obtained by TVA and MS, respectively.

| Sample Name | Samples Obtained by TVA | Samples Obtained by MS |
|---|---|---|
| Fe-B (10 at. %) | $Fe_{0.91}B_{0.0797}O_{0.01}W_{0.0003}$ | $Fe_{0.51}B_{0.045}C_{0.045}O_{0.37}H_{0.03}$ |
| Fe-B (20 at. %) | $Fe_{0.6997}B_{0.28}O_{0.02}W_{0.0003}$ | $Fe_{0.536}B_{0.13}C_{0.05}O_{0.24}H_{0.044}$ |
| Fe-B (30 at. %) | $Fe_{0.6197}B_{0.3}O_{0.07}W_{0.0003}$ | $Fe_{0.48}B_{0.2}C_{0.03}O_{0.26}H_{0.03}$ |
| Fe-B (40 at. %) | $Fe_{0.5597}B_{0.38}O_{0.05}W_{0.0003}$ | $Fe_{0.43}B_{0.255}C_{0.01}O_{0.28}H_{0.025}$ |
| Fe-B (50 at. %) | $Fe_{0.4497}B_{0.45}O_{0.05}W_{0.0003}$ | $Fe_{0.33}B_{0.3}C_{0.02}O_{0.32}H_{0.03}$ |

**Table 3.** Atomic composition of samples Fe-B obtained by TVA.

| Sample | TVA |
|---|---|
| Fe-B (10 at. %) | $Fe_{0.91}B_{0.09}$ |
| Fe-B (20 at. %) | $Fe_{0.71}B_{0.29}$ |
| Fe-B (30 at. %) | $Fe_{0.67}B_{0.33}$ |
| Fe-B (40 at. %) | $Fe_{0.60}B_{0.40}$ |
| Fe-B (50 at. %) | $Fe_{0.50}B_{0.50}$ |

It is worth noticing that the real compositions are almost similar to the nominal ones in cases with low B content (i.e., less than 10 at. %) as well as in cases with higher B content (i.e., higher than 40 at. %), whereas in cases with intermediate B content, the real compositions are higher than the nominal ones.

*3.2. Mössbauer Spectroscopy*

The Mössbauer spectra of the Fe-B thin films obtained by TVA are presented in Figure 6. The emission spectrum of sample with 5 at. % B nominal composition consists in a sextet with narrow lines and hyperfine parameters close to metallic Fe (body centered cubic structure –bcc). Hence, the specific isomer shift (IS) is 0.01 mm/s and the hyperfine magnetic field ($B_{hf}$) is 33.1 T (0.00 mm/s and 33.15 T, respectively, for metallic Fe). The corresponding quadrupole splitting (QS) is 0.00 mm/s. Two superposed magnetic components, i.e., an external narrow and an inner broad magnetic sextet, respectively, are evidenced in the spectrum of the Fe-B film with 10 at. % B nominal composition. The first magnetic pattern with IS = 0.01(1) mm/s and $B_{hf}$ = 33.1 T and with a relative spectral contribution of 29% corresponds again to a metallic Fe-like phase. The broad sextet was fitted via a hyperfine magnetic field distribution (presented on the right side of the spectrum), corresponding to a continuous distribution of local atomic configurations around the central Fe, specific to an amorphous Fe-B intermetallic phase.

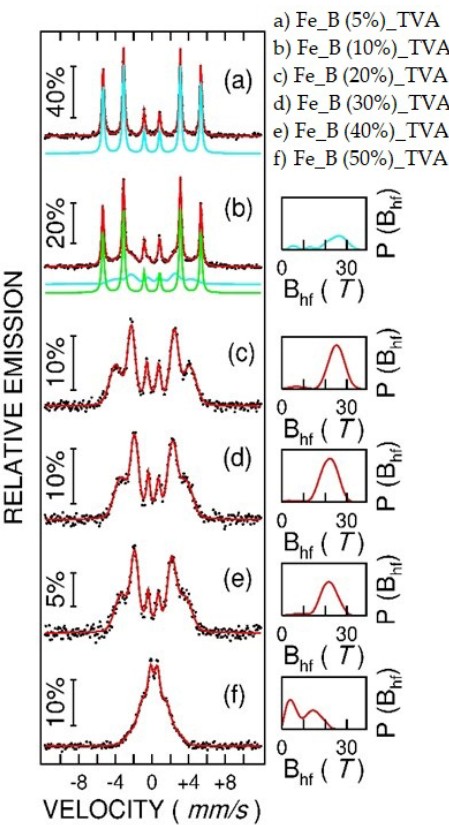

**Figure 6.** RT CEM spectra of Fe-B thin films obtained by TVA, with the following nominal compositions of B (in at. %): 5% (**a**), 10% (**b**), 20% (**c**), 30% (**d**), 40% (**e**), 50% (**f**). The vertical bars indexes the effect of the spectra (in %). The probability distributions of the hyperfine magnetic field are shown on the right hand of the spectra. The black dots are experimental points, the blue and green continuous lines provides the spectral components discussed in the text whereas the red lines represent the total spectral envelope given by the fit.

The specific average hyperfine parameters of this Fe-B intermetallic phase are: <IS> = 0.14 mm/s, <Bhf > = 24 T and QS almost 0.0 mm/s. An average of the hyperfine parameters over the two main local configurations corresponding to the two spectral components gives: 29.4 T for the average hyperfine magnetic field and 0.04 mm/s for the average Isomer Shift. Further on, at higher B contents, the Mössbauer spectra consist only in broad magnetic sextets, specific to amorphous Fe based phases [26–29], which start to collapse at higher B or RE content due to enhanced magnetic relaxation. We noted the uniform distribution of $B_{hf}$ with nominal B content between 20 and 40 at. % and the tendency of forming two average local configurations for B content of 50 at. % simultaneously with enhanced magnetic relaxation phenomena. On the other hand, the intensity ratio between the second and the third absorption line, $R_{23}$, in either the elemental sextets under the hyperfine magnetic field distributions at higher B concentrations as well as in the narrow sextet of samples of low B concentrations is 4.0, giving a direct proof for an in-plane orientation of the Fe spins [26] in the Fe-B films obtained by TVA.

The evolution of the average hyperfine magnetic field and average isomer shift versus the real B content in such intermetallic compounds in case of reduced magnetic relaxation (i.e., for B nominal content lower than 40 at. %) is given in Figure 7. An almost linear decrease in the average hyperfine magnetic field and an almost linear increase in the isomer shift is observed. Similar tendencies were also previously reported for Fe-B compounds [12].

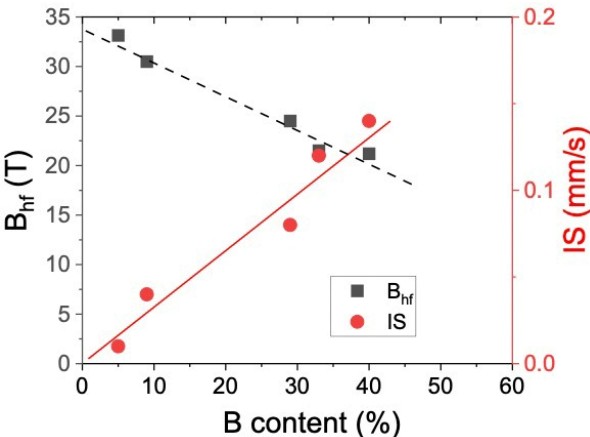

**Figure 7.** The dependence of average $B_{hf}$ (black points) and IS (red points) versus the real B content in the intermetallic Fe-B thin films obtained by TVA.

The Mössbauer spectra of Fe-B thin films obtained by MS are shown in Figure 8. Two distinct features from the RT CEM spectra of films obtained by TVA are to be mentioned in this case: (i) the presence of a relative intense central component, and (ii) the $R_{23}$ ratio between the second and the third emission line of the external sextet in the spectra of low B content is 0, giving a direct indication for a perpendicular to the film plane orientation for the Fe spins in the intermetallic compounds. In this context, the following fitting procedures were considered for the films obtained by MS. The central components were always fitted by a narrow hyperfine field distribution extending from 0 up to maximum 20 T, whereas depending on the nominal B content, either an external relative broad sextet (at low B concentration) or an external hyperfine field distribution spreading out from 10 to 35 T (at high B content) were considered. The corresponding hyperfine magnetic field distributions are shown on the right side of each spectrum.

We noted that <IS> values belonging to the central component accounted by the distribution at low fields (e.g., with $<B_{hf}>$ from 6 to 11 T, depending on the sample composition) range between 0.4 and 0.5 mm/s, which are specific values for $Fe^{3+}$ ions, whereas <IS> values belonging to the broad external sextet accounted by the distribution at high fields (e.g., with $<B_{hf}>$ from 20 to 29 T, depending on the sample composition) range between 0.1 and 0.2 mm/s, as specific to Fe-B intermetallic phases with a B content ranging from approximately 25 at. % to 45 at. % (see Figure 7). As a consequence, and also based on the RBS results presented in Table 2, the central component was assigned to very disordered $Fe_2O_3$ clusters under a strong magnetic relaxation regime. The distributed sextet of higher average hyperfine magnetic fields to an amorphous Fe-B phase of higher amount of B and the most external sextet is evidenced mainly for films of low B content, to a disordered nanosized metallic Fe phase with inclusions of B.

In cases with samples with a nominal content of B of 5 at. %, the external sextet is characterized by $B_{hf}$ = 32.3(2) T and IS = 0.01(1) mm/s, suggesting the formation of an intermetallic Fe-B phase with 7(2) at. % of B with the Fe spins oriented perpendicularly to the film plane. Its relative spectral contribution is 50(1) %, meaning that almost half of the Fe atoms enter in this phase. The rest of the Fe atoms/ions give rise to the Fe oxide clusters (50(1) % spectral contribution).

A quite similar spectral decomposition results also for the film with a nominal content of 10 at. % of B. A higher IS value for the intermetallic component (IS = 0.04(1) mm/s) suggests a higher B content in the Fe-B phase (approximately 10(2) at. % of B) of this sample. The relative spectral area of the Fe oxide phase is approximately 53(1) %, meaning that approximately 27 $Fe^{3+}$ ions and 40 $O^{-2}$ ions (from 100 atoms) participate in the Fe oxide phase and approximately 24 Fe atoms and 3 B atoms participate in the Fe-B phase, in reasonable agreement with RBS data in Table 2. The same perpendicular to plane

orientation of the Fe spins in the Fe-B phase is evidenced by Mössbauer spectroscopy ($R_{23} = 0$) for this sample as well.

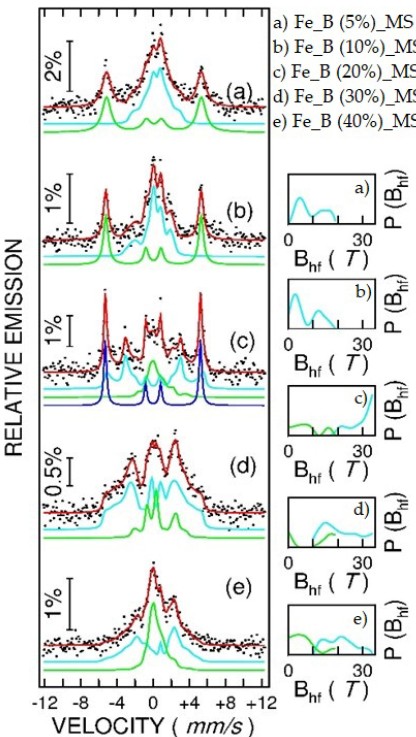

**Figure 8.** RT CEM spectra of Fe-B thin films obtained by MS, with the following nominal compositions of B (in at. %): 5% (**a**), 10% (**b**), 20% (**c**), 30% (**d**), 40% (**e**). The vertical bars indexes the effect of the spectra (in %). The probability distributions of the hyperfine magnetic field are shown on the right hand of the spectra The black dots are experimental points, the blue and green continuous lines provide the spectral components discussed in the text, whereas the red lines represent the total spectral envelope given by the fit.

In cases with samples with nominal content of B of 20 at. %, except the external sextet characterized bt $B_{hf}$ = 32.9(1) T and IS = 0.01(1) mm/s and associated to a local configuration close to nanosized metallic Fe (i.e., of very low B content), there is a second component fitted by a distribution of hyperfine magnetic fields with $<B_{hf}>$ =29.2(2) T and $<IS>$ =0.12(2) mm/s, easily associated via Figure 7 to a Fe-B phase with about 30 at. % of B (i.e., $Fe_{70}B_{30}$). The relative spectral area of the last component is 44(1) % meaning, according to Table 2, 23 atoms of Fe and 11 atoms of B (from 100 atoms) in this Fe-B phase. The most inner spectral component, fitted this time via a distribution of hyperfine magnetic fields with $<B_{hf}>$ =7.2(2) T and $<IS>$ =0.45(2) mm/s, is associated with the $Fe_2O_3$ phase. By the relative spectral area of this component of 26(1) %, a number of 14 ions of $Fe^{3+}$ and 21 atoms of $O^{2-}$ from 100 atoms contribute to the Fe oxide phase, again in reasonable agreement with the RBS data in Table 2. We mention the perpendicular to the film plane orientation of the Fe spins in the metallic phase poor in B (similar to the one of spectra corresponding to films with less than 10 at. % of B) and the in plane orientation of the Fe spins in the amorphous intermetallic phase of higher B content ($Fe_{70}B_{30}$).

Only two spectral components were considered for samples of higher B content. In cases with samples with nominal content of B of 30 at. %, the outer magnetic component, fitted via a distribution of hyperfine magnetic fields with $<B_{hf}>$ = 20.5(2) T and $<IS>$ = 0.15(2) mm/s, was assigned to an amorphous intermetallic Fe-B phase of type $Fe_{60}B_{40}$ (see Figure 7), whereas the inner magnetic component was fitted by a distribution of hyperfine magnetic fields characterized by $<B_{hf}>$ = 10.5(2) T and $<IS>$ = 0.53(2) mm/s to the $Fe_2O_3$ phase. The relative spectral areas of 70(2) % and 30(2) %, respectively, infer 33 atoms of Fe

and 22 atoms of B in the Fe-B phase and 15 ions of $Fe^{3+}$ and 22 ions of $O^{2-}$ in the Fe oxide phase from 100 atoms, again in reasonable agreement with Table 2. A similar reasoning is applied for the film with nominal composition of 40 at. % of B. Accordingly, the Mössbauer results support the formation of an amorphous intermetallic phase of type $Fe_{52}B_{48}$ ($<B_{hf}>$ = 19.5(2) T and $<IS>$ =0.19(2) mm/s) with relative spectral area of 54(2) % and an Fe oxide phase ($<B_{hf}>$ = 6.5(2) T and $<IS>$ = 0.51(2) mm/s) with a relative spectral area of 46(2) %. The corresponding atomic contribution at 100 atoms in the film is 23 Fe and 21 B in the Fe-B phase and 18 Fe and 28 O in the Fe oxide phase, in reasonable agreement with RBS results. The Fe spins of the intermetallic phases reached in B are in the film plane ($R_{23}$ = 4).

### *3.3. Magnetic Characterization*

The large capabilities of the MOKE method to investigate the magnetic behavior in such thin films, including in plane anisotropies, are illustrated by the hysteresis loops presented in Figures 9 and 10. The loops, consisting of the dependence of the Kerr angle on the applied magnetic field, were collected on the Fe-B thin films with thicknesses ranging from 70 to 100 nm, however, much thicker than the penetration depth of laser radiation (wavelength of 632 nm). Specific loops collected on films prepared by TVA, with different concentrations of B are shown in Figure 9. Three orientations of the magnetic field relative to the (110) direction of the Si substrate, 0°, 45° and 90°, were considered for each sample in order to investigate the in-plane anisotropy, as previously reported in [30,31]. Briefly, the idea is that in the case of uniaxial anisotropy, the hysteresis loops evolve from a rectangular shape, specific to a field applied along the easy axis of magnetization to a linear shape, specific to a field applied along the hard axis of magnetization (which is perpendicular to the easy axis in this case). If the saturation field coincides with the switching field the magnetization reversal is achieved by a coherent rotation of the spins (e.g., for a magnetic monodomain), whereas in the case of a saturation field higher than the switching field, the magnetization reversal is via domain wall displacements. If the evolution of the loops is not between these two limits, then we deal with an angular distribution of the easy axis (the broader the distribution, the lower the variation of the loop shape, when the film is rotated, if the loops remain constant at different angular rotations of the samples (from 0 to 90° in the present experiments) a lack of in-plane anisotropy is demonstrated.

In this context, the following observation can be drawn from Figure 9 for the films obtained by TVA: (i) the films with a B content lower or equal to 40 at. % are all ferromagnetic, (ii) the magnetic anisotropy is of uniaxial type and increases progressively with the B content starting from the case of isotropic behavior in those samples with very low B content, where mainly crystalline intermetallic phases are evidenced by Mössbauer spectroscopy (iii) magnetic domains are formed in the films and the magnetization reversal is through displacement of magnetic walls. (iv) The films are soft magnetic, with coercive fields ranging between 20 and 30 Oe, depending on the B content and reaching a maximum for the sample with 30 at. % of B which is completely amorphous and presents negligible magnetic relaxation.

The specific loops collected on films with equivalent nominal compositions of B, but prepared by MS, are shown in Figure 10. The MOKE signal of the samples with the lowest B content (i.e., with nominal compositions of B lower than 10 at. %) is more than one order of magnitude lower than in the case of the films obtained by TVA. This observation, correlated also with the linear shape of negligible coercivity of the loop, gives support for the assignment of the MOKE signal to the Fe oxide phase rather than to the Fe-B intermetallic one. Nevertheless, this assignment is also supported by the Mössbauer results, proving the perpendicular to the film plane alignment of the Fe spins in the Fe-B phase of low B content with no influence on the in-plane magnetization reversal observed by the present longitudinal MOKE geometry. We noted that this intermetallic phase with a relative amount of 50% pointed by Mössbauer spectroscopy would be the only one susceptible to contribute to the magnetization reversal if the Fe spins had in-plane components. The situation is drastically changed in cases of the films with higher nominal B content (i.e., 30

or 40 at. %) where the MOKE signal can be assigned to the Fe-B amorphous intermetallic phase of the higher B content with the Fe spins in the sample plane. This time, the MOKE signal increased by more than one order of magnitude, however, remaining still lower than in the case of films obtained by TVA of equivalent nominal B content. This is due to a relatively lower content of the intermetallic phase in the film (more than 40% of Fe oxide are formed).

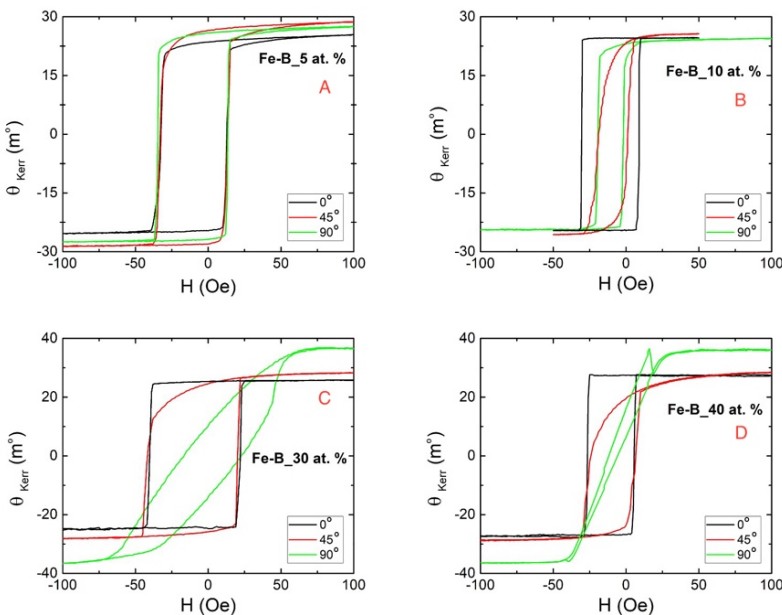

**Figure 9.** MOKE hysteresis loops for Fe-B films obtained by TVA. Different orientations of the magnetic field with respect to a reference in plane direction are considered. (**A**) is Fe-B 5 at. %, (**B**) is Fe-B 10 at. %, (**C**) is Fe-B 30 at. %, and (**D**) is Fe-B 40at. %.

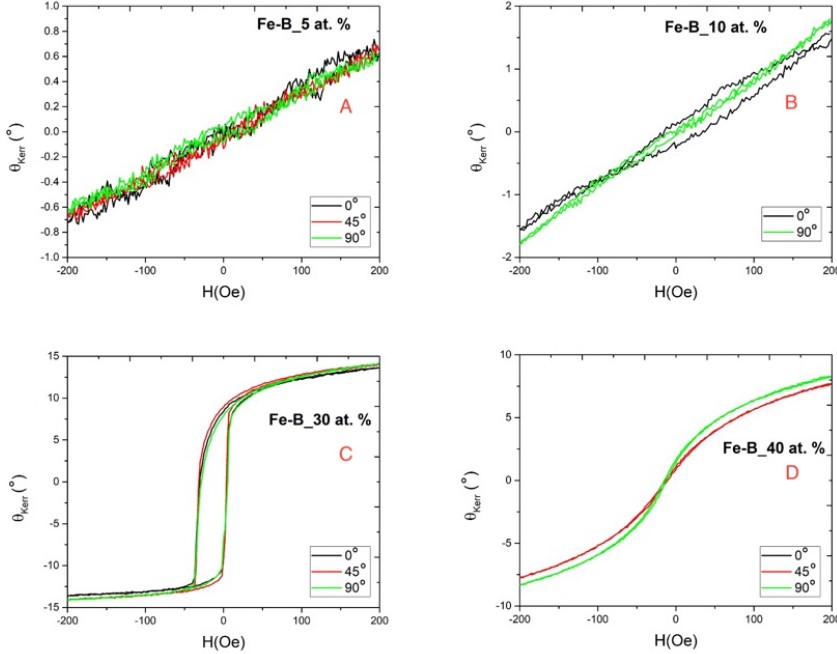

**Figure 10.** MOKE hysteresis loops for Fe-B films obtained by MS. Different orientations of the magnetic field with respect to a reference in plane direction are considered. (**A**) is Fe-B 5 at. %, (**B**) is Fe-B 10 at. %, (**C**) is Fe-B 30 at. %, and (**D**) is Fe-B 40at. %.

As a last observation, the loop of the sample with nominal content of B of 40 at. % is almost closed, with almost negligible coercivity, resembling rather a loop specific to a paramagnetic material, most probably due to the high content of B in the real $Fe_{52}B_{48}$ phase, where magnetic relaxation phenomena are substantially enhanced.

## 4. Conclusions

The paper aimed to analyze Fe-B films with various concentrations of B (5 at. % −50 at. %) in terms of morpho-structural and magnetic properties. Thin films (70–100 nm) were successfully obtained by two deposition methods, TVA and MS. The morpho-structural investigations performed by SEM, XRD and XRR have shown a roughness-free morphology of all samples, but with a much better uniformity of samples obtained by TVA. The elemental analysis was surveyed by EDX for samples of high B content and performed in detail by RBS based techniques. The formation of relative pure Fe-B intermetallic films by TVA and a high content of oxygen in films obtained by MS were proven. The real phase composition of the films was obtained by Mössbauer Spectroscopy. The obtained results on the elemental composition are in reasonable agreement with the RBS data. Mössbauer results give evidence for no Fe oxide phase in the films obtained by TVA but the formation of disordered clusters of $Fe_2O_3$ in the films obtained by MS, in a relative amount between 20 to 50 %, depending on the B content. It was proven that in the films of low B content (less than 10 at. % of B nominal composition), mainly a defect crystalline phase approaching the body centered cubic structure of metallic Fe with B inclusions is formed whereas at increased B concentrations, an amorphous Fe-B phase of high B content is formed. The Fe spins are always oriented in the sample plane for the films obtained by TVA whereas in the case of the films obtained by MS, the Fe spins are perpendicular to the sample plane for the crystalline-like phase and in the film plane in the case of the amorphous-like intermetallic compounds. These results give direct explanation for the different magnetic behavior of the films, as evidenced by the longitudinal MOKE investigations. The highest magnetic anisotropy and saturation magnetization, as well as the lowest coercivity, is obtained for the $Fe_{60}B_{40}$ amorphous films obtained by TVA. The formation of nano-crystalline Fe-B films with perpendicular to plane magnetic anisotropy by MS is another direction of interest for magneto-functional applications.

**Author Contributions:** Conceptualization, C.S. and V.K.; methodology, C.S. and V.K.; validation V.K. and C.P.; investigation, C.S., P.D., B.B., I.B., C.L., A.A.D. and O.G.P.; data curation, V.K., C.S., I.B., C.L., A.A.D. and O.G.P.; writing—original draft preparation, C.S.; writing—review and editing, V.K. and C.P.; Visualization, C.S.; Supervision, C.P., V.K. and C.P.L.; Project administration; C.P.L., V.K. and C.P.; Funding acquisition, C.P.L., V.K. and C.P. All authors have read and agreed to the published version of the manuscript.

**Funding:** This work was supported by grants of the Romanian Ministry of Education and research, Core Program PN-19 (Contract No. 21N/2019) and project Supported Institutional Excellence Contract 35 PFE/2022 at NIMP and the Core Program LAPLAS IV—contract 16N/2019 at NILPRP. The research leading to these results has also received funding from the NO-RO grants 2014–2021, under Project contract No 39/2021 and the POC Project REBMAT, ID P_37_697. Ion beam experiments were performed at 3 MV Tandetron accelerator from NIPNE-HH and were supported by the Romanian National Program "Instalații și Obiective de Interes Național" (Grant No. 786/2014).

**Institutional Review Board Statement:** Not applicable.

**Informed Consent Statement:** Not applicable.

**Data Availability Statement:** Not applicable.

**Conflicts of Interest:** The authors declare no conflict of interest.

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
