# Peer review of "Structural and Magnetic Specificities of Fe-B Thin Films Obtained by Thermionic Vacuum Arc and Magnetron Sputtering"

_coatings, doi:10.3390/coatings12101592_

Round 1

Reviewer 1 Report

Please find the referee report in the attached PDF file.

Author Response

Response to reviewer 1

Manuscript reference:  Coatings-1964474

Dear Editor,

Dear Reviewer,  

Please find enclosed the revised version of our manuscript entitled “Structural and Magnetic Specificities of Fe-B Thin Films obtained by Thermionic Vacuum Arc and Magnetron Sputtering” by Cornel Staicu, Claudiu Locovei, Alexandru Dinu, Ion Burducea, Paul Dinca, Bogdan Butoi, Oana Pompilian, Corneliu Porosnicu, Cristian Petrica Lungu and Victor Kuncser, (Ref No 1964474) submitted for publication in the special issue “Advanced of Nanoparticles and Thin Films” in Coatings. 

We thank the Reviewers for their critical assessments and constructive comments of our work, on the basis of which we have revised and corrected our manuscript. We addressed all reviewers concerns and below you find a point-by-point response to all the questions and concerns raised by the two referees. The revisions to the manuscript were marked up using the “Track Changes” function of MS Word and they were subsequently written with red. 

We hope that in its improved form, the reviewers and you will find our manuscript suitable for publication in the special issue of Coatings.

Sincerely yours,

Dr. Victor Kuncser  

Author's Reply to the Review Report (Reviewer 1)

We thank the Reviewer for the critical considerations. We hope that by addressing them we increased the quality of the revised manuscript in order to be accepted for publication.

1.   “In the Results and discussion section, the authors presented three different aspects of the Fe-B film samples, i.e., morpho structure (3.1), elemental composition (3.1 & 3.2), and magnetic characterization (3.3); and then the authors discussed the potential magnetic-functional applications primarily based on the magnetic characterizations. I’m wondering if there are any direct relationships between the aforementioned three aspects, especially how does the different morpho structures and the elemental compositions affects the magnetic structures for TVA and MS samples? I believe with this information, the authors can make the entire study more comprehensive and complete, and to form a more straightforward storyline, i.e., morpho structure/elemental composition → magnetic structure → magnetic applications.”

Response: Many thanks to the reviewer for this observation. The idea of the above-mentioned correlations was also our aim with this paper and therefore the entire structure of the paper was built in this respect. As a consequence, the last sub-section 3.3 has discussed the magnetic behavior of the samples as observed by MOKE in relation to the morpho-structural and elemental analysis investigated in the first two sub-sections 3.1 & 3.2 (please see the discussions from lines 431 to 439, where it is explained the reason for a very low magnetization in case of samples of low B content obtained by MS, the much higher magnetization in samples of higher B content, however lower than in similar samples obtained by TVA, and the very low coercivity in samples with highest B content). However, we felt more comfortable with an additional explanation in this part, introduced at lines 440-442, as follows: “To note that this intermetallic phase with a relative amount of 50% pointed by MÓ§ssbauer spectroscopy would be the only one susceptible to contribute to the magnetization reversal if the Fe spins would have in-plane components.” Some additional correlations were also added between lines 417-423 in the part describing the magnetic properties of samples obtained by TVA, as follows: ”(ii) the magnetic anisotropy is of uniaxial type and increases progressively with the B content starting from the case of isotropic behavior ”in those samples with very low B content, where mainly crystalline intermetallic phases are evidenced by MÓ§ssbauer spectroscopy” (iii) magnetic domains are formed in the films and the magnetization reversal is through displacement of magnetic walls. (iv) the films are soft magnetic, with coercive fields ranging between 20 and 30 Oe, depending on the B content ”and reaching a maximum for the sample with 30 at. % of B which is completely amorphous and presents negligible magnetic relaxation.”   

      The same type of correlations structure-elemental composition-magnetic properties were also mentioned in the last part of the conclusions.       

2. “In Table 1, the Fe/B effective thickness does not change proportional with the B concentration? Is this related to the elemental analysis in section 3.1 (Table 2 & 3)? Can the authors further elaborate on this? Can the authors explain what is the difference between the nominal compositions (Table 2) and the real compositions (Table 3), especially how did the authors obtain the real composition values in Table 3?”

Response: The ratio of the Fe/B effective thicknesses would change proportional with the composition only in case of very similar weights of the two elements. In our case, the weight of B is order of magnitude lower than of Fe and hence the effective thickness of Fe is order of magnitude higher than for B. In order to double the composition of B is enough to almost double the effective thickness of B, decreasing very slowly the effective thickness of Fe for keeping the same total thickness of the film (keeping constant the Fe effective thickness, we will keep the right composition, but the total thickness of the film will increase). Concerning the second part of the observation, the difference between Table 2 and Table 3, is that table 2 include all the elements whereas Table 3 is reported only to the renormalized composition of the Fe-B intermetallic phase (the atomic composition is 100 % on only Fe+B). 

3. “As I understand, the authors prepared 6 different B concentrations for each of the TVA and MS methods. However, in different measurements, only some of the B concentrations were used. For instance, in Figure 6, all the 6 concentrations were plotted, while in Figure 8, only 5 out of the 6 concentrations were plotted. Can the authors explain the rationale behind the different selections of the concentrations for different measurements? From my opinion, it will make the manuscript better arranged if the authors can use the same complete set of B concentrations for different measurements.”

Response: For sure, it would be nice to have reliable measurements on all the considered samples. However, in the above-mentioned case of MÓ§ssbasuer results, it was not possible to collect reliable spectra on both samples (obtained by MS and TVA) with 50 at % of B. That because the samples obtained by TVA were partially enriched in 57Fe whereas the ones obtained by MS are only with natural Fe (therefore the effects in the TVA samples are one order of magnitude higher than in those obtained by MS). The sample with the highest content of B is the one with the lowest content of Fe, leading finally in case of the unenriched sample obtained by MS to a CEM spectrum of very poor statistics, where a paramagnetic pattern can be only guessed, not undoubtable evidenced.  

4. “All the figures in this manuscript have extremely low resolutions. It is very difficult for the referees to read these figures. Therefore, can the authors update these figures with higher resolutions?”

Response: The resolution of the figures was improved (the new ones were inserted and the old ones were deleted by track changes) anywhere was possible in the manuscript (except RBS representations and MÓ§ssbauer results, where the data are plotted in specific graphical programs providing finally jpg files with 600 dpi resolution, which however is quite reasonable) 

5. “For figures with subplots (or multiple panels), e.g., Figure 1, 2, 4, 5, 6, 8, can the authors add proper annotation to each subplot, like which synthesis methods is used (TVA or MS), which B concentration it is? It can make these figures much easier to read.”

Response: The suggested annotations were introduced in the new figures. 

6. “For Figure 6, can the authors add descriptions in the figure caption to explain what the black dots and multiple lines with different colors represent? And the left panel has 6 different B concentrations, however, for the probability distributions on the right panels, there are only 5 subplots, can the authors elaborate on this?”

      Response: The captions of Figs. 6 and 8 were completed according to the suggestion by: The black dots are experimental points, the blue and green continuous lines provides the spectral components discussed in the text whereas the red lines represent the total spectral envelope given by the fit.

      The lack of the probability distribution associated to the spectrum of sample with 5 at. % of B is the fact that this spectrum was fitted only by a crystalline sextet (unique value for the hyperfine magnetic field corresponding to the unique position of Fe in the bcc structure of metallic Fe). Therefore, the notion of hyperfine field distribution (typical to the amorphous state) is meaningless for this sample.  

7. “For Figure 7, the authors used the inset to plot the evolution of the average IS. I’m wondering if it is better to plot IS in a separate panel or overplot the IS on the same panel with a second y axis (on the right side)?”

      Response: Figure 7 was modified according to the reviewer’s suggestion (with 2 y axis) and the caption was changed accordingly.

Reviewer 2 Report

The authors obtained interesting results on the features of the structural, morphological and magnetic properties of Fe-B thin films obtained by different methods. It has been proven the formation of relative pure Fe-B intermetallic films by TVA and a high content of oxygen in films obtained by MS.

Unfortunately, the article contains many inaccuracies and errors.

When describing the experiment, it is not indicated how the EDX measurements were carried out.

• In the text of the article there is a link to Figure 5 (lines 192-193) where the results of EDX are given, but there is no figure itself.

• Lines 176-177 swapped references to red and black line

• Lines 187-188- It is not clear the explanation about the different thickness of the films obtained by two methods

• lines 211-212 are confused a) and b) in the caption to fig.

• Lines 213-214 of Table 2 – Fe concentration values in the Fe-B(20%) sample are less than in the Fe-B(30%) sample, if this is true, then this should be explained in the text of the article.

• Lines 236–237 the phrase does not match what we see in the table

• It is not clear from the text of the article whether the MÓ§ssbauer spectra were obtained on the same samples for which the structural and magnetic properties were studied.

• The quality of Figures 6 and 7 needs to be improved

• Authors should carefully proofread the text of the article, check the data in the tables and improve the figure.

Author Response

Response to reviewer 2

Manuscript reference:  Coatings-1964474

Dear Editor,

Dear Reviewer,  

Please find enclosed the revised version of our manuscript entitled “Structural and Magnetic Specificities of Fe-B Thin Films obtained by Thermionic Vacuum Arc and Magnetron Sputtering” by Cornel Staicu, Claudiu Locovei, Alexandru Dinu, Ion Burducea, Paul Dincă, Bogdan Butoi, Oana Pompilian, Corneliu Porosnicu, Cristian Petrica Lungu and Victor Kuncser, (Ref No 1964474) submitted for publication in the special issue “Advanced of Nanoparticles and Thin Films” in Coatings.

We thank the Reviewers for their critical assessments and constructive comments of our work, on the basis of which we have revised and corrected our manuscript. We addressed all reviewers concerns and below you find a point-by-point response to all the questions and concerns raised by the two referees. The revisions to the manuscript were marked up using the “Track Changes” function of MS Word and they were subsequently written in red.

We hope that in its improved form, the reviewers and you will find our manuscript suitable for publication in the special issue of Coatings.

Sincerely yours,

Dr. Victor Kuncser 

Author's Reply to the Review Report (Reviewer 2)

We thank the Reviewer for his useful comments that helped us improve the presentation of our work.

  1. “When describing the experiment, it is not indicated how the EDX measurements were carried out.”

Response: The description of the EDX measurement was introduced in the revised manuscript (lines 135-140) as follows: ”Energy-dispersive X-ray spectroscopy (EDX) was carried out for topographic inspection and local chemical analysis. Typically, it is impossible to detect B by EDX. However, O X-ray yields are easily identified. The INCA package software was used for the EDX elemental analysis. The energy calibration of the EDX facility was carried out with the Kα X-ray emissions of copper (8.05 keV) and of silicon (1.74 keV).”

  1. “In the text of the article there is a link to Figure 5 (lines 192-193) where the results of EDX are given, but there is no figure itself.”

Response: Sorry for our typing mistake related to an abandoned less representative Figure (due to the increased number of Figures in the manuscript). The text in the revised manuscript (pag 6 lines 200-203) was changed accordingly:”The EDX measurements were made” on the two films with 50 at. % of B obtained by TVA and MS, respectively.

  1. “Lines 176-177 swapped references to red and black line”

Response: Thank you for mentioning this new mistake, we have modified both the graph (Fig.3) and the text (line 185-186) in the revised manuscript.

  1. “Lines 187-188- It is not clear the explanation about the different thickness of the films obtained by two methods”

Response: The geometry in the two devices (and especially the position of the microbalance with respect to the sample place) is different and therefore some deviations (usually higher in TVA) could appear between the nominal (desired) thickness and the real one. This is the reason that the thickness of a film inside the same series obtained by a technique is usually checked by XRR or Optical interferometry.

  1. “lines 211-212 are confused a) and b) in the caption to fig.”

Response: RBS and RNBS data are obtained on the samples and an example of each method is shown in Figure 5. It is not clear to us the remark of the reviewer.  

  1. “Lines 213-214 of Table 2 – Fe concentration values in the Fe-B(20%) sample are less than in the Fe-B(30%) sample, if this is true, then this should be explained in the text of the article.”

Response: Thank you again for your valuable observation. Unfortunately, we have done a mistake in the initial manuscript by changing the compositions provided by NRBS in between the samples with nominal composition of 20 at % and 30 at. % of Boron, respectively. Now we have proceeded to this inversion in Tables 2 and 3, by mentioning the correct compositions in the revised manuscript. Also the discussion of point 7 is correct now. 

  1. “Lines 236–237 the phrase does not match what we see in the table”

Response: Please see the answer at point 6 in the frame of the corrected Table 3.

  1. “It is not clear from the text of the article whether the MÓ§ssbauer spectra were obtained on the same samples for which the structural and magnetic properties were studied.”

Response: Yes, the MÓ§ssbauer data were obtained on the same samples where the structural and magnetic data were obtained. Firstly, the structural (XRD and XRR) investigations were performed and then the MÓ§ssbauer ones. Finally, a smaller piece was cut from the sample previously investigated. This smaller piece was further investigated by MOKE measurements.

  1. “The quality of Figures 6 and 7 needs to be improved”

Response: Figure 7 (page 9 line 309) was modified with a better representation with a much better quality. In case of figure 6, the data are plotted in specific graphical programs providing finally jpg files with 600 dpi resolution, which however is quite reasonable).

  1. “Authors should carefully proofread the text of the article, check the data in the tables and improve the figure.”

Response: We double checked the manuscript for typos or any other types of mistakes, everything is now modified with track changes in the article submitted final form. The resolution of the figures was improved (the new ones were inserted, and the old ones were deleted by track changes) anywhere was possible (except RBS representations and MÓ§ssbauer results, where the data are plotted in specific graphical programs providing finally .jpg files with 600 dpi resolution, which however is quite reasonable).

Round 2

Reviewer 1 Report

The authors have properly responded to the comments left by the referee and made corresponding adjustments to the manuscript. With this, I think now the results presented in the paper are sound and the quality of the research meets the publication standards of Coatings, therefore, I would recommend the acceptance for publication.

Reviewer 2 Report

Accept in present form